# Indoor Air Pollutant (Toluene) Reduction Based on Ultraviolet-A Irradiance and Changes in the Reactor Volume in a TiO_2_ Photocatalyst Reactor

**DOI:** 10.3390/ma16196399

**Published:** 2023-09-25

**Authors:** Yong-Woo Song, Seong-Eun Kim, Min-Sang Yoo, Jin-Chul Park

**Affiliations:** 1School of Architecture and Building Science, Chung-Ang University, Seoul 06794, Republic of Korea; 2Graduate School, Chung-Ang University, Seoul 06794, Republic of Korea; asteria03@naver.com (S.-E.K.); gkgk0911@naver.com (M.-S.Y.)

**Keywords:** TiO_2_ photocatalyst, toluene, ultraviolet-A (UV-A) irradiance, reactor volume, reduction trend equation

## Abstract

This study experimentally confirmed the effect of TiO_2_ photocatalysts on the removal of indoor air pollutants. In the experiment, toluene, a representative indoor air pollutant, was removed using a coating agent containing TiO_2_ photocatalysts. Conditions proposed by the International Organization for Standardization (ISO) were applied mutatis mutandis, and a photoreactor for an experiment was manufactured. The experiment was divided into two categories. The first experiment was conducted under ISO conditions using the TiO_2_ photocatalyst coating agent. In the second experiment, the amount of ultraviolet-A (UV-A) light was varied depending on the lamp’s service life, and the volume of the reactor was varied depending on the number of contaminants. The results showed that the TiO_2_ photocatalytic coating agent reduced the effect of toluene. This reduction effect can be increased as a primary function depending on the changes in the amount of UV-A light and reactor volume. However, because toluene is decomposed in this study, additional organic pollutants such as benzene and butadiene can be produced. Because these pollutants are decomposed by the TiO_2_ photocatalysts, the overall reduction performance may change. Nonetheless, TiO_2_ photocatalysts can be used to examine the effect of indoor pollutant reduction in indoor ventilation systems and building materials.

## 1. Introduction

In modern living conditions, humans spend over 80% of their day indoors [1]. Therefore, it is necessary to decrease indoor air pollutants to avoid adverse effects on human health. The global COVID-19 pandemic has raised awareness of the importance of indoor environments, and the Center for Disease Control and Prevention recommends improving indoor air environment conditions through ventilation and the removal of pollutants [2]. In general, ventilation is insufficient in buildings that are designed to save energy due to airtightness [3,4,5,6,7].

Therefore, methods need to be developed for removing indoor air pollutants and mitigating their impact. The representative pollutants affecting the indoor air environment are particulate matter and gaseous pollutants, such as carbon dioxide and toluene. Compared to particulate pollutants in the air that settle on the floor over time, gaseous pollutants come into contact with the bronchus and skin through respiration, which can adversely affect human health even at low concentrations. Therefore, it is important to reduce gaseous pollutants in indoor environments [8].

Among the different indoor air pollutants, volatile organic compounds (VOCs) can be significantly harmful to the human body when their indoor concentration exceeds 0.3 mg/m^3^ [9]. Consequently, many institutions classify them as pollutants that need to be managed [10,11,12].

According to a survey conducted on pollutant status conducted for new apartments in Seoul, Republic of Korea, from 2015 to 2020, toluene constituted an average of approximately 55% of the total VOCs. Therefore, the impact of VOCs can be reduced by removing toluene.

One of the methods for reducing toluene is using titanium dioxide (TiO_2_) photocatalysts [13,14,15,16,17], which generates OH and O_2_^−^ radicals with strong oxidizing power on the surface through photochemical reactions with ultraviolet (UV) light, given as [18,19,20,21,22,23,24,25]:
Photoexcitation: $TiO_{2}+hv\;\to \;h^{+}+e^{-}$(1)Oxidation reaction: $OH^{-}+h^{+}\;\to \;OH^{c}$(2)Reduction reaction: $O_{2\left(ads\right)}+e^{-}\;\to \;O_{2\left(ads\right)}^{-}$(3)Ionization of water: $H_{2}O\;\to \;OH^{-}+H^{+}$(4)Protonation of superoxide: $O_{2}^{-}+H^{+}\;\to \;HOO^{\cdot }$(5)Electron scavenger: $HOO^{\cdot }+e^{-}\;\to \;HOO^{-}$(6)Formation of H_2_O_2_: $HOO^{-}+H^{+}\;\to \;H_{2}O_{2}$(7)$OH\;+pollution+\;O_{2}\cdot^{-}\;\to \left(CO_{2},\;H_{2}O,\;etc.\right)$,(8)
where $h^{+}$ and $e^{-}$ are the hole and electron, respectively, $OH^{c}$ is the hydroxyl radical, and hv is UV.

According to the corresponding reaction formulas, toluene can be changed into a material with low molecular bonding through the reduction of TiO_2_ photocatalyst. In particular, byproducts such as benzene, butadiene, and ethylbenzene may be produced. Toluene, as a VOC-based substance, can have some adverse effects on the human body such as allergies and headaches. However, TiO_2_ photocatalysts are expected to have a low impact on the entire human body and the environment because they can decompose the organic matter in VOCs.

Existing studies on the application of TiO_2_ photocatalysts to construction were primarily conducted on building materials to reduce specific pollutants.

Ohko et al. [26] investigated whether a thin coating of TiO_2_ photocatalysts on type 304 stainless steel (SUS 304) could impart corrosion resistance and preventive performance. Janus et al. [27] exposed TiO_2_-coated concrete plates to artificial sunlight and found that TiO_2_ coatings reduced the activation of *E. coli*, thereby providing antibacterial properties. Kolarik and Toftum [28] lowered the concentration of indoor air pollutants emitted from chipboards and carpets by conducting photochemical reactions using UV/visible-light bulbs installed in indoor spaces that used cement-based paint mixed with TiO_2_ photocatalysts. Luna et al. [29] studied the self-cleaning and depolluting properties of TiO_2_ photocatalysts on the surface of in situ building materials.

Several studies were conducted to confirm specific pollutant reduction. Hüsken, Hunger, and Brewers [30] analyzed the air purification performance through nitric oxide (NO) reduction using concrete products containing TiO_2_ photocatalysts. Ramirez et al. [31] analyzed the porosity properties of concrete surfaces and confirmed that TiO_2_ photocatalytic coatings on concrete reduced toluene levels. Mentecchio et al. [32] designed a reaction device and applied it to computational fluid dynamics simulations, confirming that TiO_2_ reduced VOCs and gaseous pollutants. Zhang et al. [33] demonstrated that, when doped photocatalysts are applied on a glass plate, the formaldehyde (HCHO) levels in indoor environments decrease by over 20% compared to ventilation alone.

A literature survey shows that TiO_2_ photocatalysts, coating agents, and paints have primarily been used in building materials, while NOx and toluene have been investigated as target pollutants to confirm the effectiveness of TiO_2_ photocatalysts [26,29,34,35]. Shen et al. [36] and Li et al. [37] conducted experiments to improve the performance of the decomposition of toluene, a representative indoor pollutant, by doping TiO_2_ photocatalysts with metallic substances. Mo et al. [38] and Tomašić et al. [39] demonstrated that toluene is decomposed through TiO_2_ photocatalysis, resulting in organic pollutants such as benzene and ethylbenzene, which are further decomposed by the TiO_2_ photocatalysts. Some studies were conducted in outdoor spaces where UV rays were utilized to exploit the bandgap energy (3.8 eV) of TiO_2_ photocatalysts; visible light was studied by doping TiO_2_ photocatalysts for indoor application [40,41,42,43,44]. However, fewer airflow elements are present in indoor environments, and the pollutant reduction efficiency and activity of TiO_2_ photocatalysts are lower than that of those with UV rays.

The aforementioned studies focus on indoor and outdoor building materials, and some additional considerations must be taken when considering indoor ventilation systems. First, when a photocatalyst is applied inside the device and an artificial light source is added, the efficiency of the photocatalyst may vary depending on the intensity of the light source. Song et al. [45] demonstrated that applying UV rays in confined spaces such as ventilation ducts, as well as preventing the rays from leaking indoors (to avoid UV damage to all life forms), can safely remove indoor air particulate matter. However, only a few studies have investigated changes in UV-A irradiance according to the lifespan of artificial light sources. The second thing to consider is the distance between the photocatalyst coating installed in a narrow device and the light source. Depending on this distance, the performance with regards to decomposition of pollutants may change; however, very few studies have investigated this.

Therefore, as a basic research step towards utilizing photocatalysts in indoor ventilation systems, this study aims to examine the reduction in UV-A irradiance and the pollutant removal performance according to the changes in reactor volume caused by varying the distance between the photocatalyst coating and the artificial light source. This study is expected to help suggest considerations for performance variables when using photocatalysts in indoor ventilation systems in the future.

## 2. Experimental Method and Materials

### 2.1. Overview

This experiment is significant as a basic research step to determine the appropriate amount of light and duct diameter when using photocatalysts in ventilation devices. First, the light intensity of lamps installed in ventilation devices decreases depending on their lifespan, which may also reduce pollutant reduction performance. In general, the lifespan of a lamp is considered to be until the initial light output decreases by 20 to 30% [46]. Accordingly, in this experiment, the toluene reduction concentration was determined based on when the lamp’s light output dropped by 25% compared to before and when it dropped to 50%, which is an even worse condition. Second, the performance review according to volume change aims to examine the impact of changes in ventilation duct size. A change in volume occurs depending on the distance between the light source and the photocatalyst coating; accordingly, the corresponding toluene reduction concentration was determined.

A TiO_2_ photocatalytic coating agent was used to examine its toluene-reduction performance. Some of the conditions suggested in ISO 22197 [47] (UV-A irradiance of 10 W/m^2^, pollutant flow rate of 3 lpm and concentration of 1.00 ppm, type of light-transmitting glass, etc.) were utilized for the experiment (International Organization for Standardization, 2011).

The ISO basic conditions summarized in Table 1 were used for the experimental conditions. The output of these conditions was reflected in the photoreactor, as shown in Figure 1 and Figure 2.

The photoreactor was made of aluminum and a wire mesh was inserted into the reactor to form a constant airflow. Figure 1 and Figure 2 show the schematic and photographs of the photoreactor prepared and installed, respectively. In addition, to minimize the influence of light other than that from the photoreactor and UV lamp used in the experiment, the space where the experiment was conducted was configured as a dark room.

The test piece included in the photoreactor used Pyrex glass (Schott Korea Co. Ltd., Seoul, Republic of Korea), which does not affect the photocatalytic reaction. An 18 W black light lamp (Philips, Seoul, Republic of Korea) that utilizes the UV-A wavelength band (320–400 nm) was used to produce UV light for the experiment. Figure 3 shows the power spectrum provided by the lamp manufacturer [48]. The wavelength range of the lamp can be determined through the power spectrum, and irradiance of approximately 20 W/m^2^ is generated. Quartz glass was used for the photoreactor as proposed in ISO 22197-3 [47] to ensure the UV-A light reached the test piece smoothly. The glass can transmit 90% or more of the UV-A light (SCHOTT) [49].

### 2.2. Experimental Method and Equipment

Figure 4 and Table 2 present the experimental setup for reducing toluene by using the prepared photoreactor. The TiO_2_ photocatalyst comprises measurement equipment and a test gas.

To confirm the toluene reduction effect of the TiO_2_ photocatalyst, the following information was applied to the experiment. Toluene gas in a cylinder compressed to 100 atm pressure was used, and nitrogen was assumed to be added to the carrier gas at a concentration defined in the carrier gas, as specified in the analysis certificate provided by the gas supplier. Next, mixed gas containing pollutants with a flow rate of 3.0 L/min was injected into the photoreactor through wet air mixing using a ball flow meter and a regulator to constantly maintain the measured flow rate of 1.5 L/min, as listed in Table 2. The remaining mixed gas with a flow rate of 1.5 L/min was discharged via the bypass.

Additionally, the reaction effect of the TiO_2_ photocatalyst can significantly vary depending on the relative humidity (RH) level. Therefore, the RH of toluene gas during the experiment was measured using the bypass section, and a 50% RH level was confirmed.

The measurement equipment can measure the concentration of toluene used at 10 s intervals by using a photoionization detector (PID), which measures the VOC system components. Because the VOC system is toluene gas in this case, the concentration recognized by the measuring equipment can be estimated as the concentration of toluene.

### 2.3. Materials

The TiO_2_ photocatalyst coating agent (LT-01, from Bentech Frontier, Jeollanam-do, Republic of Korea) was used in liquid form, and approximately 18 g of the material was applied to a sample holder and a glass specimen (UV-A irradiance and volume of reactor change experiment) through spray coating. The spray was applied using a spray gun as a general painting method, and the application was confirmed by measuring the weight before and after application. The sample was then dried and used in the experiment. The weight of the applied coating agent was estimated by measuring the weight of the specimen before and after coating. Table 3 summarizes the composition of the coating agent phase, as per the material safety data sheets (MSDSs). In addition, according to the contents of the coating agent provided by MSDSs, toxicity (skin corrosiveness or irritability, severe eye damage or irritability, skin irritability, etc.) was evaluated using rabbits and guinea pigs, and it was judged to be low.

The material constituting the coating agent was identified through field emission SEM (FE-SEM) and EDS mapping analysis (see Table 4) at the Chonnam National University Joint Lab (Yeosu City, Jeollanam-do, Republic of Korea). This analysis was used by Rengui et al. [50] and Nguyen et al. [51] to study the properties of the TiO_2_ photocatalyst and coating agent used in this study.

### 2.4. Experimental Conditions

Table 5 summarizes the experimental conditions for toluene reduction by the TiO_2_ photocatalytic coating agent according to ISO 22197 [47]. The experiment was performed at least three times for each condition to account for variance owing to experimental errors.

To prevent damage to the TiO_2_ photocatalyst coating agent applied to the experimental specimen considering the repeated experiments, the experimental specimen was stored separately in a vial.

The initial UV-A irradiance was 20 W/m^2^, which decreased by 25% (15 W/m^2^) and 50% (10 W/m^2^) owing to the reduction in irradiance depending on the service life of the lamp. The UV-A irradiance values were consistent with the level during winters in South Korea (13.4 W/m^2^) based on the comprehensive information statistics generated by the climate change monitoring efforts of the Korean Meteorological Administration [52]. In addition, the temperature and humidity values applied in the experiment were similar to those of apartment complexes in Korea, as confirmed through measurements by Hwang et al. [53], assuming that actual indoor temperature and humidity values can be used.

This irradiance level is an average value measured on the horizontal surface, and higher values were also considered. In this case, the irradiance was changed by adjusting the voltage of the Slidac transformers, as summarized in Table 6.

Additionally, the volume of the reactor was changed from 100% to 80% and 60%, as summarized in Table 7. As the volume decreased, the distance between the ISO photoreactor and the UV-A lamp decreased, which was confirmed by the changes in the volume of the pollutants. Furthermore, the combination with construction facility systems such as ducts and heat exchangers was also considered (and is discussed in the next section).

## 3. Results

### 3.1. FE-SEM and EDS Mapping Results

FE-SEM and EDS mapping analyses of the coating agent were conducted to confirm the effect of toluene reduction on using the TiO_2_ photocatalyst coating agent. Table 8 and Figure 5 present the results of the component ratio analysis of the TiO_2_ photocatalyst coating agent.

Figure 5a illustrates the result of the FE-SEM and EDS mapping experiment before the TiO_2_ coating agent is applied, and Figure 5b illustrates the measurement result of the experiment piece after the TiO_2_ coating agent is applied.

Before applying the TiO_2_ coating, aluminum (81.24%) exhibited the highest compositional proportion. After coating, titanium (0%→51.35%) exhibited the highest proportion. Furthermore, the proportion of oxygen increased (4.81%→45.42%), which confirmed the presence of the TiO_2_ component in the coating agent. Therefore, the toluene reduction effect of the TiO_2_ photocatalyst can be confirmed when the coating agent is used in the experiment.

### 3.2. Condition 1 (UV-A Irradiance 10 W/m^2^, Volume Change)

The experiment was conducted by utilizing a UV-A irradiance of 10 W/m^2^ and changing the volume of pollutant by applying the ISO experimental conditions. The results are shown in Figure 6 and listed in Table 9.

As summarized in Table 9, the concentration of toluene decreased by approximately 8% depending on the change in the photoreactor volume (from 100% to 60%), which increased from 20.79% to 27.38% owing to the operation of the UV-A lamp as the gas passed through the photoreactor. Furthermore, the decomposition efficiency of the TiO_2_ photocatalyst improved owing to the changes in the volume of the reactor as the amount of toluene inside the photoreactor decreased.

### 3.3. Condition 2 (UV-A Irradiance 15 W/m^2^, Volume Change)

Figure 7 and Table 10 present the experimental results obtained by utilizing a UV-A irradiance of 15 W/m^2^, representing a 60% increase compared to that in condition 1, as well as changing the pollutant volume to confirm the changes in the toluene reduction performance depending on the increase in UV-A irradiance.

The experimental results showed that when 100% reactor volume was applied, the toluene concentration decreased by 0.23 ppm (22.55%), which increased the reduction effect by approximately 2% compared to that under ISO conditions (10 W/m^2^, 100% volume of reactor). This effect can be attributed to the difference in the injection concentration. To further confirm the performance of toluene reduction owing to the changes in the amount of light and the volume of the reactor, the experiment was conducted for different irradiance values and reactor volumes. The results showed that when the volume of the reactor was reduced to 60%, the reduction in concentration increased to 0.29 ppm (28.71%). Therefore, the decomposition efficiency was improved owing to the reduction in the amount of toluene that moved inside the photoreactor, which is similar to the previous results for the changes in the volume of the reactor.

### 3.4. Condition 3 (UV-A Irradiance 20 W/m^2^, Volume Change)

Figure 8 and Table 11 present the results when the maximum UV-A lamp irradiance of 20 W/m^2^ was applied and the pollutant volume was changed.

The toluene concentration decreased by 0.25 ppm (24.75%) when the 100% volume of the reactor was applied, resulting in a 4% increased reduction effect compared with that under the ISO conditions.

Additionally, the reduction in concentration increased to 0.28 ppm (27.45%) and 0.32 ppm (32.00%) when 80% and 60% volumes of the reactor were applied, respectively. As seen in the previous experiment, the reduction efficiency increased as the oxidation reaction increased, and the amount of toluene decreased as the irradiance increased. This indicated that the toluene reduction rate by the TiO_2_ photocatalyst can be increased by increasing the irradiance or decreasing the volume of the reactor.

## 4. Discussion

This study applied a TiO_2_ photocatalytic coating agent under the previously stated ISO experimental conditions to analyze its toluene reduction effect. Figure 9 shows the combined results of the toluene-reduction trend experiments that applied UV-A irradiance and the volume of the reactor as variables.

In Figure 9, section ① exhibited the highest reduction rate of 31.69% when the UV-A irradiance was 20 W/m^2^, and the volume of the reactor was 60%. In section ②, the toluene concentration decreased, and the reduction rate was higher and lower than 25% and 30% (minimum and maximum of 24.99% and 28.97%), respectively. The reduction effect can be obtained even at low UV-A irradiance by decreasing the volume of the reactor. The effect can also be achieved by increasing the UV-A irradiance as the volume of the reactor increases. In section ③, the toluene concentration decreased from 20.84% to 24.11%. In this section, we discuss how a high reduction rate can be achieved when the UV-A irradiance and volume of the reactor are changed simultaneously.

The experimental results showed that when the UV-A irradiance and volume of the reactor were changed, the reduction rate increased by over 1.5 times in sections ①, ②, and ③. Accordingly, the amount of toluene removed in each condition was calculated using the removal-amount calculation formula mentioned in ISO 22197-3 [47]. Table 12 lists the formulas used for the calculation (ISO 22197-3:2013 [47]), and Table 13 lists the calculated removal amounts.

The standard deviation of the removal amount by condition was 0.02–0.04, as summarized in Table 13; therefore, all data are expected to be distributed around the average.

Figure 10 shows the results of the reduction trend owing to UV-A irradiance and the volume of the reactor, which were obtained from the toluene removal amounts calculated in Table 13.

The trend equations of the graphs shown in Figure 9 can be calculated as listed in Table 14. The slope of the graph increased almost linearly as the UV-A irradiance increased under a constant volume of the reactor. Furthermore, the y-intercept increased as the volume of the reactor decreased. The toluene reduction rate increased by 27% when 60% volume of the reactor was applied instead of 100% volume.

However, in the case of the TiO_2_ photocatalyst, the efficiency was assumed to be lower than that of the doped photocatalyst considering it is a pure photocatalyst material [54]. Therefore, this study confirmed the toluene reduction performance of TiO_2_ photocatalysts by applying specific variables that were suggested by previous studies. The findings indicate that applying a doped photocatalyst at a later stage can yield a higher effect compared to the reduction efficiency and amount of reduction achieved in this study.

Therefore, a substantial toluene reduction effect can be achieved when the TiO_2_ photocatalyst is applied to building materials and air circulation devices.

This study has several key goals and implications, as well as some limitations.

First, we sought to evaluate the effectiveness of TiO_2_ photocatalytic coating in reducing the toluene concentration present indoors. This study confirms that TiO_2_ photocatalyst is effective in reducing gaseous pollutants present indoors. In addition, during a total of 27 experiments conducted in this study, the pollutant reduction effect of the coating agent is not reduced; accordingly, performance is maintained with regards to stability and durability, assuming no physical damage.

Second, although a standardized ISO experimental method is used in this study, the real environment is difficult to represent. In particular, in indoor environments, the amount of contaminants can vary compared to experimental conditions; moreover, UV-A light levels can vary. By experimentally confirming these variables, this study not only confirms the efficacy of TiO_2_ photocatalyst, but also provides basic research results that confirm the benefits of using TiO_2_ photocatalyst coatings for reducing indoor pollution.

However, additional research is needed to confirm the presence of UV-A indoors and its effectiveness at very low concentrations. In addition, additional measurements such as GC and FT-IR are required for the by-products produced by the decomposition of materials with high molecular bonding such as toluene. The incorporation of more diverse variables is an avenue for possible future research. For instance, the effect of doped TiO_2_ confirmed through previous studies can be combined with the variables applied in this study to further enhance the TiO_2_ photocatalytic effect indoors.

As a result, this study provides meaningful parameters and results for the utilization of TiO_2_ photocatalyst for indoor toluene abatement.

## 5. Conclusions

This study confirmed the removal of toluene, an indoor air pollutant, by the application of a TiO_2_ photocatalyst coating. Most previous studies applied TiO_2_ photocatalysts to building materials such as concrete and paint and dealt with NO-concentration reduction using external UV light. However, with the recent increase in indoor living time, the importance of indoor air quality has been further emphasized, and we conducted an experimental study using TiO_2_ photocatalyst coatings to confirm that TiO_2_ photocatalyst reduces the concentration of toluene, a representative indoor pollutant. The results of this study can be summarized as follows.

First, an ISO-based standard method for reducing pollutants in TiO_2_ photocatalysts was applied for the experiment, and the internal properties of the TiO_2_ photocatalyst coating used in the ISO TiO_2_ photoreactor experiment were analyzed using FE-SEM and EDS mapping analysis. The analysis examined the content of the internal constituent material according to the application of the coating agent to FE-SEM and EDS mapping samples; evidently, the Ti content in the coating agent was 51.29%, which was suitable for conducting an experiment on toluene reduction.

Second, because of the experiment under the ISO experimental conditions, the application of the TiO_2_ coating agent reduced to 20.79% (1.01 to 0.80 ppm) (UV-A: 10 W/m^2^, reactor volume: 25 cm^3^). Furthermore, toluene concentration reduced within 22.55% (0.23 ppm) to 32.00% (0.32 ppm) depending on UV-A irradiation intensity and reactor volume (UV-A: 15 and 20 W/m^2^, reactor volume: 20 and 15 cm^3^). These results confirmed that the reduction effect can be increased by about 1.5 times compared to the case of using basic ISO conditions.

Third, using the experimental results, trend equations were derived to identify the decreasing trends based on UV-A irradiation illumination and reactor volume. Evidently, all trend equations showed almost the same slope (0.05–0.055). In addition, the value of R^2^ was higher than 0.9. Therefore, it was inferred that the degree of toluene reduction by the photocatalyst coating agent was proportional to the change in reactor volume. Accordingly, the performance with regards to reducing pollutants is evidently improved as the number of times pollutants come into contact with the photocatalyst application site increases, even indoors.

The results of this study confirmed that toluene can be reduced by changing the amount of UV-A light or the volume of the reactor using a coating agent with TiO_2_ photocatalysts.

However, the concentration of pollutants used in this study is higher than the actual environment. Moreover, it is difficult to obtain UV-A light indoors; therefore, there are some limitations to the experimental conditions.

Further research is needed in the future to validate the results of this study under real-world conditions and investigate the potential impact of the spray application method on coating uniformity and contaminant removal efficacy. Additionally, the review of the by-products resulting from the photocatalytic reaction is limited. According to Mo et al. [38], some unwanted by-products that are harmful to health are produced, but because their concentrations are low, the negative effects on the human body are negligible. However, continuous operation of indoor ventilation systems may have negative effects; accordingly, this will need to be examined in future research.

Despite these limitations, this study provides valuable insights into the promising potential of TiO_2_ photocatalytic coatings for reducing toluene in real indoor environments through UV-A irradiation and reactor volume tuning.

## Figures and Tables

**Figure 1 materials-16-06399-f001:**
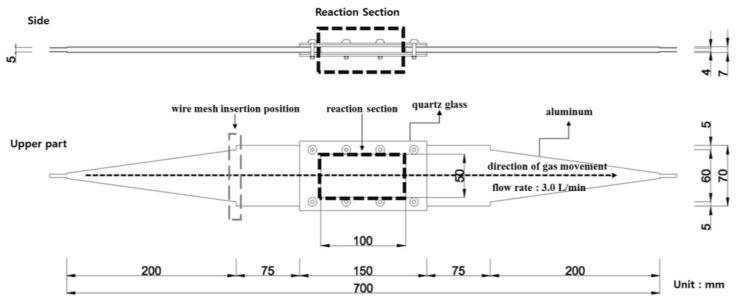
Conceptual diagram of the photoreactor.

**Figure 2 materials-16-06399-f002:**
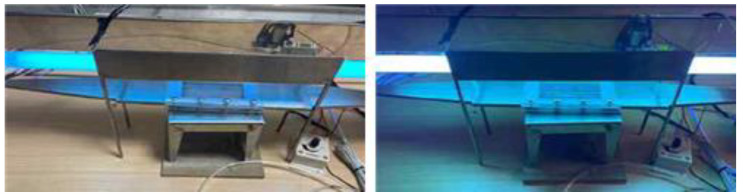
Photograph of the photoreactor installation.

**Figure 3 materials-16-06399-f003:**
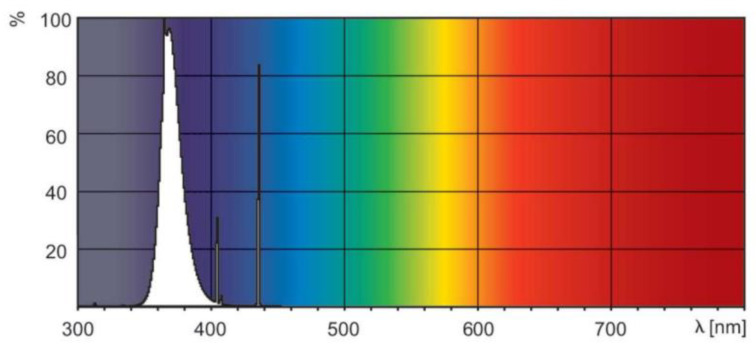
Spectral power distribution.

**Figure 4 materials-16-06399-f004:**
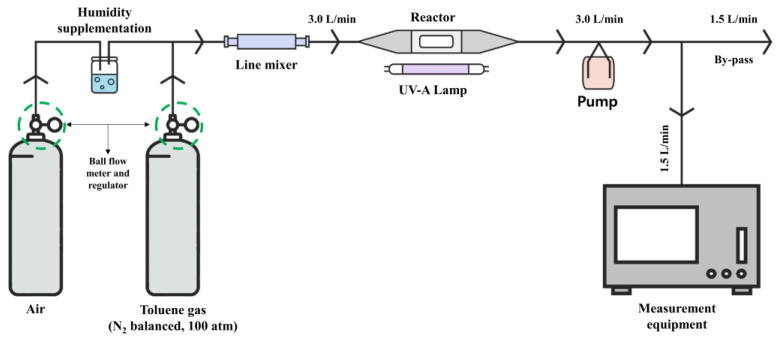
Experimental setup.

**Figure 5 materials-16-06399-f005:**
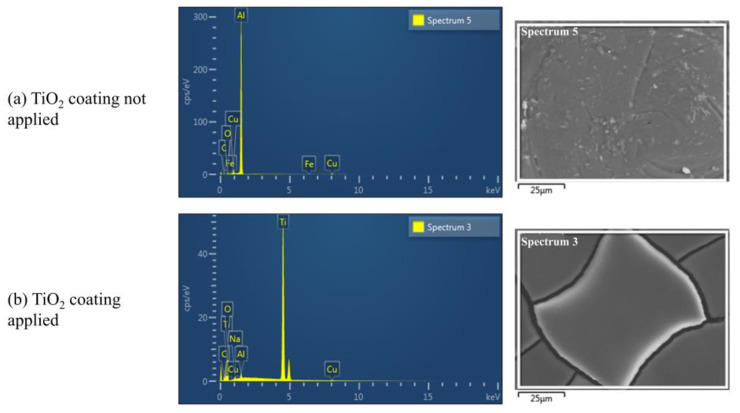
FE-SEM and EDS analysis results of the TiO_2_ coating agent material identification.

**Figure 6 materials-16-06399-f006:**
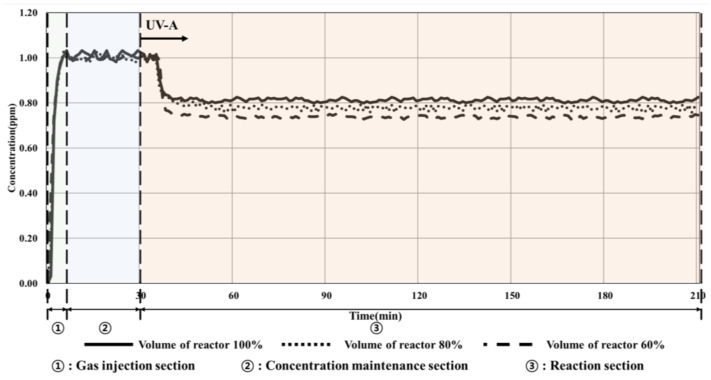
Condition 1 (10 W/m^2^, volume change) experimental results.

**Figure 7 materials-16-06399-f007:**
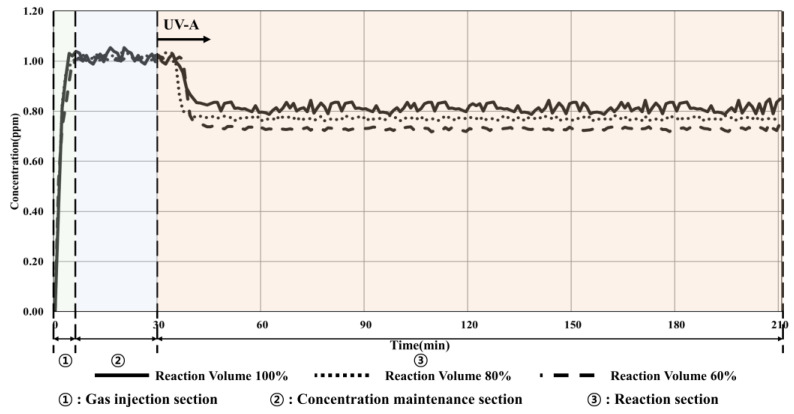
Condition 2 (15 W/m^2^, volume change) experimental results.

**Figure 8 materials-16-06399-f008:**
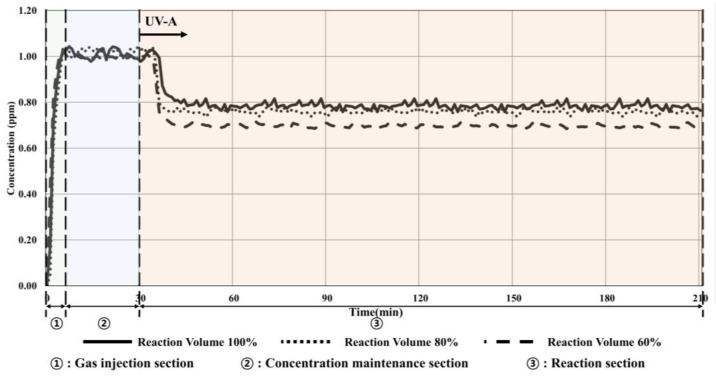
Condition 3 (20 W/m^2^, volume change) experimental results.

**Figure 9 materials-16-06399-f009:**
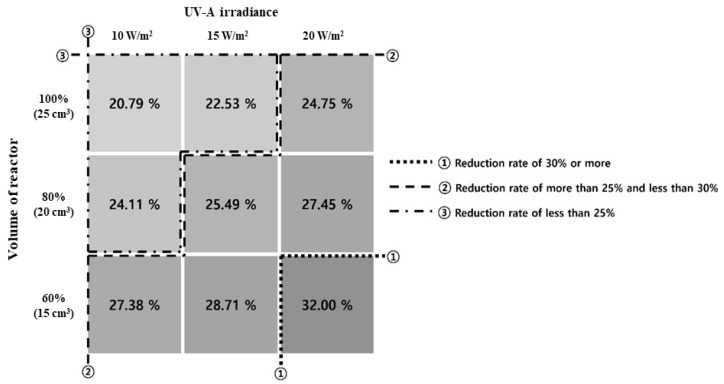
Comprehensive matrix of results of toluene reduction reactor experiments.

**Figure 10 materials-16-06399-f010:**
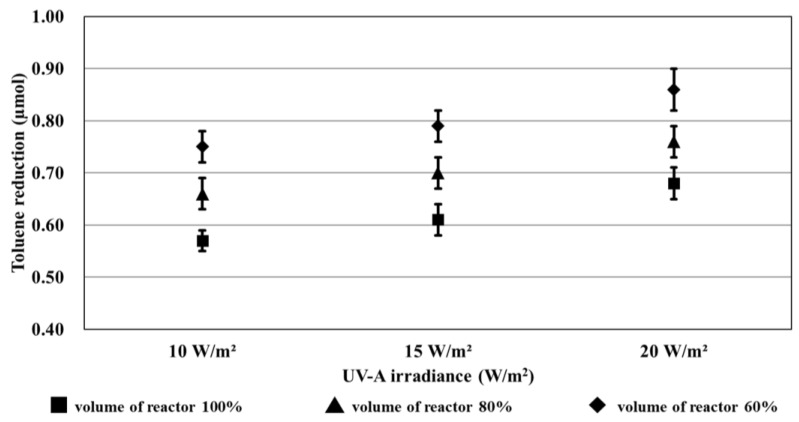
Results of the reduction trend owing to UV-A light irradiance and volume of the reactor.

**Table 1 materials-16-06399-t001:** Basic conditions for ISO temperature, humidity, and photoreactors.

Basic Temperature	Basic Humidity	Airtight Optical Window Glass Material
25 °C ± 2.5 °C	RH 50%	Quartz or borosilicate glass
**Test piece length (l_1_)**	**Test piece width (l_2_)**	**Air layer thickness (l_g_)**
99.9 mm ± 1.0 mm	49.9 mm ± 1.0 mm	5.0 mm ± 0.5 mm

**Table 2 materials-16-06399-t002:** Experimental measurement equipment.

Equipment	Specification
Gas analyzer 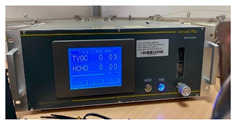	Model	Airwell plus (KINSCO Technology, Seoul, Republic of Korea)
Measuring gas	Toluene
Sensor type	PID sensor
Measurement range	0–10 ppm
Measured flow rate	1.5 L/min
UV light meter	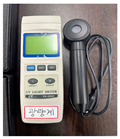	Model	YK-35UV (Lutron, Lutron Headquarters & Lighting Control Institute7200 Suter Road, Coopersburg, PA, USA)
Measurement ranges and resolution	Range 1: 2 mW/cm^2^1.999 mW/cm^2^ × 0.001 mW/cm^2^Range 2: 20 mW/cm^2^19.99 mW/cm^2^ × 0.01 mW/cm^2^
Accuracy	±(4% FS + 2 dgt)
Temperature & Humidity Data Logger	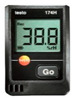	Model	174H (Testo Korea, Seoul, Republic of Korea)
Temp. measurement range	−20 to +70 °C
%RH measurement range	0–100% RH

**Table 3 materials-16-06399-t003:** TiO_2_ coating composition stated in MSDSs.

Contained Chemicals	Proportion
TiO_2_ (anatase)	3.4%
Isobutanal	0.1%
Sodium hydroxide	0.1%
Water	96.4%

**Table 4 materials-16-06399-t004:** FE-SEM and EDS mapping measurement equipment.

Classification	Contents	
Model	SIGMA 500 (Carl Zeiss)	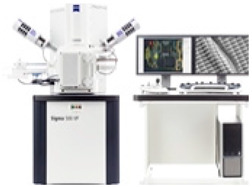
Detector	SE 2	
EDS detector	X-Max^N^50 (Oxford)	
Acceleration voltage	18.0 kV	
Working distance	8.5 mm	
Magnification	500× to 1000×	
Time resolution	0.8 nm	

**Table 5 materials-16-06399-t005:** Experimental conditions for toluene reduction.

Experiment Condition	UV-A Irradiance	Volume of Reactor	Gas Concentration	Temperature	Humidity	Time
Condition 1 (volume change)	10.0 W/m^2^	100% (25 cm^3^)	1.00 ppm	25 ± 2.5 °C	RH 50%	3 h
		80% (20 cm^3^)				
		60% (15 cm^3^)				
Condition 2 (UV-A irradiance 50% up and volume change)	15.0 W/m^2^	100% (25 cm^3^)				
		80% (20 cm^3^)				
		60% (15 cm^3^)				
Condition 3 (UV-A irradiance 100% up and volume change)	20.0 W/m^2^	100% (25 cm^3^)				
		80% (20 cm^3^)				
		60% (15 cm^3^)				

※ Number of experiments: at least three times for each condition.

**Table 6 materials-16-06399-t006:** UV-A irradiance based on voltage.

Voltage (V)	UV-A Irradiance (W/m^2^)	Experimental Application
75	2.5	
80	3.5	
90	4.0	
95	5.5	
100	7.0	
110	7.5	
120	8.5	
125	10.0	○
130	11.0	
140	13.0	
150	15.0	○
160	16.0	
170	17.0	
180	17.5	
190	18.5	
200	20.0	○
Not applied Slidac	19.5	

**Table 7 materials-16-06399-t007:** Experimental volume of the reactor.

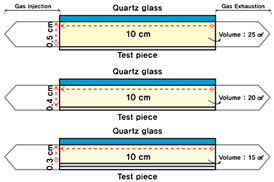	**Content**	**W × D × H**
	Standard volume (100%)	25 cm^3^ (10 cm × 5 cm × 0.5 cm)
	Reduced volume (80%)	20 cm^3^ (10 cm × 5 cm × 0.4 cm)
	Reduced volume (60%)	15 cm^3^ (10 cm × 5 cm × 0.3 cm)

**Table 8 materials-16-06399-t008:** Results of the compositional material ratio of the TiO_2_ photocatalytic coating material.

Element	Before TiO_2_ Coating (wt%)	After TiO_2_ Coating (wt%)	Change in Value (%)
C	8.22	0.73	−7.49
O	4.81	45.66	+40.85
Na	-	1.04	+1.04
Al	81.24	0.68	−80.56
Fe	0.58	-	−0.58
Cu	5.15	0.60	−4.55
Ti	-	51.29	+51.29
Total	100.00	100.00	0

※ After the TiO_2_ coating, the proportion of O increases by 40.85%, from 4.81% to 45.66%, and the proportion of Ti increases from 0% to 51.29%. ※ Approximately 18 g of the coating agent is applied to the sample holder through spray coating, which is then dried and used in the experiment.

**Table 9 materials-16-06399-t009:** Experimental results for condition 1.

Content	Start Concentration	End Concentration	Reduction in Concentration	Reduction Rate
UV-A irradiance 10 W/m^2^, volume of the reactor 100%	1.01 ppm	0.80 ppm	0.21 ppm	20.79%
UV-A irradiance 10 W/m^2^, volume of the reactor 80%	1.00 ppm	0.76 ppm	0.24 ppm	24.11%
UV-A irradiance 10 W/m^2^, volume of the reactor 60%	1.00 ppm	0.73 ppm	0.27 ppm	27.38%

**Table 10 materials-16-06399-t010:** Experimental results for condition 2.

Content	Start Concentration	End Concentration	Reduction in Concentration	Reduction Rate
UV-A irradiance 15 W/m^2^, volume of the reactor 100%	1.02 ppm	0.79 ppm	0.23 ppm	22.55%
UV-A irradiance 15 W/m^2^, volume of the reactor 80%	1.02 ppm	0.76 ppm	0.26 ppm	25.49%
UV-A irradiance 15 W/m^2^, volume of the reactor 60%	1.01 ppm	0.72 ppm	0.29 ppm	28.71%

**Table 11 materials-16-06399-t011:** Experimental results for condition 3.

Content	Start Concentration	End Concentration	Reduction in Concentration	Reduction Rate
UV-A irradiance 20 W/m^2^, volume of the reactor 100%	1.01 ppm	0.76 ppm	0.25 ppm	24.75%
UV-A irradiance 20 W/m^2^, volume of the reactor 80%	1.02 ppm	0.74 ppm	0.28 ppm	27.45%
UV-A irradiance 20 W/m^2^, volume of the reactor 60%	1.00 ppm	0.68 ppm	0.32 ppm	32.00%

**Table 12 materials-16-06399-t012:** Calculation of toluene removal using international standards for experimental methods (Equations (9) and (10)).

Calculation Formulas for Toluene Removal by the TiO_2_ Photocatalyst	
$R=\frac{\Phi_{T0}-\Phi_{T}}{\Phi_{T0}}\times 100$	Equation (9)
$n_{T}=R\times \frac{\Phi_{T0}\times f\times 60}{100\times 22.4}$	Equation (10)
$R:\;Removal\;percentage\;of\;toluene\;\left(\%\right)$ $\Phi_{T0}:\;Toluene\;supply\;volume\;fraction\;\left(\mu L/L\right)$ $\Phi_{T}:Toluene\;volume\;fraction\;at\;the\;exit\;\left(\mu L/L\right)$ $n_{T}:\;Quantity\;of\;toluene\;removed\;by\;test\;specimen\;\left(\mu \mathrm{mol}\right)$ *f: Test gas flow rate converted to standard conditions* (0.5 L/min, 0 °C, 101.3 kPa)	

**Table 13 materials-16-06399-t013:** Results of the removal of toluene by the TiO_2_ photocatalyst.

UV-A Irradiance	10.0 W/m^2^	15.0 W/m^2^	20.0 W/m^2^
Volume of the reactor 100%	0.57 µmol	0.61 μmol	0.68 μmol
Standard deviation	0.02 µmol	0.03 µmol	0.03 µmol
Volume of the reactor 80%	0.66 μmol	0.70 μmol	0.76 μmol
Standard deviation	0.03 µmol	0.03 µmol	0.03 µmol
Volume of the reactor 60%	0.75 μmol	0.79 μmol	0.86 μmol
Standard deviation	0.03 µmol	0.03 µmol	0.04 µmol

**Table 14 materials-16-06399-t014:** Reduction-trend equations based on UV-A irradiance and the volume of toluene.

Volume of Reactor	Linear Exponential Function	R^2^
100%	y = 0.055x + 0.51	0.9758
80%	y = 0.05x + 0.6067	0.9868
60%	y = 0.055x + 0.69	0.9758

## Data Availability

Data available on request from the authors.

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
