# Peer review of "Indoor Air Pollutant (Toluene) Reduction Based on Ultraviolet-A Irradiance and Changes in the Reactor Volume in a TiO2 Photocatalyst Reactor"

_materials, 2023, doi:10.3390/ma16196399_

Round 1

Reviewer 1 Report

materials-2595131

Indoor air pollutant (toluene) reduction based on ultraviolet-A irradiance and changes in the reactor volume in a TiO2 photocatalyst reactor

Dear Authors

Your study presents a promising investigation into the application of TiO2 photocatalysts for indoor air pollutant reduction. However, there are several key areas that require substantial revision to enhance the clarity, depth, and scientific rigor of your research.

Comments:

1.       What is the underlying mechanism behind the reduction of toluene by TiO2 photocatalysts? Are there specific intermediate compounds formed during the process?

2.       How does the presence of additional organic pollutants, like benzene and butadiene, impact the overall reduction performance when toluene is decomposed?

3.       Elaborate on the potential environmental implications of decomposing toluene into other organic pollutants? How might these byproducts be controlled or mitigated?

4.       Are there any potential health risks associated with the byproducts generated from the decomposition of toluene using TiO2 photocatalysts?

5.       How do the results of this study compare with previous research on indoor air pollutant reduction using other photocatalytic materials or techniques?

6.       Can the improved reduction efficiency observed with changes in UV-A irradiance and reactor volume be attributed solely to increased photocatalytic activity, or are there other contributing factors?

7.       Have the potential long-term stability and durability of the TiO2 photocatalyst coatings been considered? How might real-world conditions affect their performance over time?

8.       Are there any indications that the TiO2 photocatalytic coating might have unintended effects on indoor air quality, such as the release of nanoparticles or other byproducts?

9.       Can authors provide insights into the potential practical applications of these findings in real-world indoor environments, such as homes, offices, or industrial settings?

10.   What are the key factors that might limit the scalability of using TiO2 photocatalyst coatings for widespread indoor air pollution reduction?

11.   Could authors elaborate on the specific methods used to ensure the uniform application of the TiO2 photocatalytic coating agent on the substrates? How was the coating thickness controlled and characterized?

12.   Given the potential for byproduct formation, what analytical techniques were employed to identify and quantify the secondary organic compounds generated during the toluene decomposition process?

13.   How were the UV-A light intensity and photoreactor volume precisely controlled and measured? Were there any challenges in maintaining consistent experimental conditions?

14.   Can authors provide more details about the photoreactor design and construction, particularly in terms of preventing any potential interference from external factors such as ambient light?

15.   Were there any challenges or limitations in accurately determining the toluene concentration levels at different stages of the experiments, especially when dealing with trace amounts?

16.   Considering that indoor environments can vary greatly in terms of temperature and humidity, how confident are authors that the observed results can be directly applied to different real-world scenarios?

17.   Did authors observe any potential saturation effects in terms of the photocatalytic capacity of the TiO2 coatings, especially under high concentrations of toluene or extended exposure times?

18.   What strategies were employed to minimize the potential loss of TiO2 photocatalyst particles from the coating during the experimental setup and operation of the photoreactor?

19.   Given the relationship between UV-A irradiance and reactor volume on the reduction efficiency, do authors envision any challenges in optimizing both parameters simultaneously for practical applications?

20.   Could authors discuss the potential challenges of scaling up the application of TiO2 photocatalyst coatings to larger surfaces, such as entire rooms or buildings, and how the results might translate to such scenarios?

Author Response

Response to Reviewers

  1. Article Title: Indoor air pollutant (toluene) reduction according to ultraviolet-A irradiance and volume of reactor change in TiO2 photocatalyst reactor
  2. Submission Date: 21 August 2023
  3. Revisions and Comments: Reviewer 1

Thank you for taking the time to review our work and providing your valuable feedback. We appreciate your thoughtful consideration and kind response. We have carefully incorporated your suggestions into the revised manuscript, making sure to reflect your opinions as much as possible. To make it easier to identify the changes, we have marked them in red.

Reviewer’s Comment

Dear Authors

Your study presents a promising investigation into the application of TiO2 photocatalysts for indoor air pollutant reduction. However, there are several key areas that require substantial revision to enhance the clarity, depth, and scientific rigor of your research.

Reviewer’s Comment

  1. What is the underlying mechanism behind the reduction of toluene by TiO2 photocatalysts? Are there specific intermediate compounds formed during the process?

Revisions and Comments

  • Thank you for these thoughtful questions. The basic reduction formula has been modified and added. During the process, benzene and ethylbenzene, which have a lower number of molecules than toluene, may be generated as intermediate compounds; however, based on previous research, these substances can also be reportedly decomposed by TiO2

Reviewer’s Comment

  1. How does the presence of additional organic pollutants, like benzene and butadiene, impact the overall reduction performance when toluene is decomposed?
  2. Elaborate on the potential environmental implications of decomposing toluene into other organic pollutants? How might these byproducts be controlled or mitigated?
  3. Are there any potential health risks associated with the byproducts generated from the decomposition of toluene using TiO2 photocatalysts?

Revisions and Comments

2, 3, 4) Thank you for these insightful questions; their answers are detailed next. Information regarding the health effects of toluene decomposition into other organic pollutants or by-products has been added to the Introduction section.

“According to the corresponding reaction formulas toluene can be changed into a material with low molecular bonding through the reduction of TiO2 photocatalyst. In particular, by-products such as benzene, butadiene, and ethylbenzene may be produced. Toluene, as a VOC-based substance, can have some adverse effects on the human body such as allergies and headaches. However, TiO2 photocatalysts are expected to have a low impact on the entire human body and the environment because they can decompose the organic matter in VOCs.”

However, the production and impact of the by-products could not be directly confirmed through this experimental study that focused on confirming the reduction in toluene. Based on previous research, the generated pollutants can also be decomposed by TiO2 photocatalyst.

Reviewer’s Comment

  1. How do the results of this study compare with previous research on indoor air pollutant reduction using other photocatalytic materials or techniques?

Revisions and Comments

  • Thank you for the pertinent question. In other studies, experiments such as metal doping were conducted to improve the reduction efficiency of photocatalysts. This study was conducted to confirm the trend of increased reduction of toluene through TiO2 photocatalyst by changing the amount of UV-A light and changing the volume of the photoreactor, aiming to reduce indoor pollutants.

Reviewer’s Comment

  1. Can the improved reduction efficiency observed with changes in UV-A irradiance and reactor volume be attributed solely to increased photocatalytic activity, or are there other contributing factors?

Revisions and Comments

  • Thank you for these thoughtful questions. Since the temperature and humidity suggested by ISO test standards were applied and the flow rate was applied consistently, only changes in UV-A light intensity and reactor volume evidently contributed to the photocatalytic activity. It is believed that there are no other contributing factors. Further information about ISO test conditions has been added to the text.

“Some of the conditions suggested in ISO 22197 (UV-A irradiance of 10W/m2, pollutant flow rate of 3 lpm and concentration of 1.00 ppm, type of light-transmitting glass, etc.)…”

Reviewer’s Comment

  1. Have the potential long-term stability and durability of the TiO2 photocatalyst coatings been considered? How might real-world conditions affect their performance over time?

Revisions and Comments

  • Thank you for these important questions. As mentioned in the text, the experiment was conducted three times for each condition, for a total of 27 experiments. During the experiments, no difference was evident in the stability and durability of the photocatalyst as no physical substances other than contaminants came into contact with the photocatalyst surface. Accordingly, further relevant text has been added to the Discussion section.

“This study confirms that TiO2 photocatalyst is effective in reducing gaseous pollutants present indoors. In addition, during a total of 27 experiments conducted in this study, the pollutant reduction effect of the coating agent is not reduced; accordingly, performance is maintained with regards to stability and durability, assuming no physical damage.”

Reviewer’s Comment

  1. Are there any indications that the TiO2 photocatalytic coating might have unintended effects on indoor air quality, such as the release of nanoparticles or other byproducts?

Revisions and Comments

  • Thank you for these thoughtful questions. This study was conducted to confirm the effectiveness with regards to reducing toluene, an indoor pollutant, and did not address the release of by-products. However, according to the analysis results on the MSDS provided by the coating material manufacturer, the photocatalytic nanoparticles are either below the standard or do not pose any hazard to the human body. This information has been added to the text.

“In addition, according to the contents of the coating agent provided by MSDSs, toxicity (skin corrosiveness or irritability, severe eye damage or irritability, skin irritability, etc.) was evaluated using rabbits and guinea pigs, and it is judged to be low.”

Reviewer’s Comment

  1. Can authors provide insights into the potential practical applications of these findings in real-world indoor environments, such as homes, offices, or industrial settings?

Revisions and Comments

  • Thank you for the insightful question. Although there are limitations to the direct application of findings to actual indoor environments, multiple insights related to the use of research results have been added to the Discussion and Conclusion sections as limitations and implications.

“However, additional research is needed to confirm the presence of UV-A indoors and its effectiveness at very low concentrations. In addition, additional measurements such as GC and FT-IR are required for the by-products produced by the decomposition of materials with high molecular bonding such as toluene. The incorporation of more diverse variables is an avenue for possible future research. For instance, the effect of doped TiO2 confirmed through previous studies can be combined with the variables applied in this study to further enhance the TiO2 photocatalytic effect indoors.”

Reviewer’s Comment

  1. What are the key factors that might limit the scalability of using TiO2 photocatalyst coatings for widespread indoor air pollution reduction?

Revisions and Comments

  • Thank you for the pertinent question. A limitation to the use of TiO2 photocatalyst indoors is the presence of UV-A indoors. The discussion regarding the limitations and need for further research was added to the Discussion section.

“Second, although a standardized ISO experimental method is used in this study, the real environment is difficult to represent. In particular, in indoor environments, the amount of contaminants can vary compared to experimental conditions; moreover, UV-A light levels can vary. By experimentally confirming these variables, this study not only confirms the efficacy of TiO2 photocatalyst, but also provides basic research results that confirm the benefits of using TiO2 photocatalyst coatings for reducing indoor pollution.

However, additional research is needed to confirm the presence of UV-A indoors and its effectiveness at very low concentrations. In addition, additional measurements such as GC and FT-IR are required for the by-products produced by the decomposition of materials with high molecular bonding such as toluene. The incorporation of more diverse variables is an avenue for possible future research. For instance, the effect of doped TiO2 confirmed through previous studies can be combined with the variables applied in this study to further enhance the TiO2 photocatalytic effect indoors.

Reviewer’s Comment

  1. Could authors elaborate on the specific methods used to ensure the uniform application of the TiO2 photocatalytic coating agent on the substrates? How was the coating thickness controlled and characterized?

Revisions and Comments

  • Thank you for these thoughtful questions. A photocatalyst coating was applied to the test piece using a spray. This method is similar to the general spray-painting method. The thickness of the coating could not be measured; moreover, the tabulated amount was calculated by measuring the weight of the test piece before and after application. The relevant discussion has been added to the text.

“The spray was applied using a spray gun as in a general painting method, and the application was confirmed by measuring weight before and after application.”

Reviewer’s Comment

  1. Given the potential for byproduct formation, what analytical techniques were employed to identify and quantify the secondary organic compounds generated during the toluene decomposition process?

Revisions and Comments

  • Thank you for the pertinent question. This study focused on confirming the reduction in toluene, and the characteristics of the by-products were not explored. These limitations have been added to the text so that they can be investigated in future research.

Reviewer’s Comment

  1. How were the UV-A light intensity and photoreactor volume precisely controlled and measured? Were there any challenges in maintaining consistent experimental conditions?

Revisions and Comments

  • Thank you for these insightful questions. For the UV-A irradiance, the voltage was controlled through the Slidac used in the experiment. Additionally, the amount of UV light emitted from the UV-A lamp was measured for each experiment. In the case of the photoreactor volume, the volume was adjusted using a glass material that prevented photochemical reactions inside the reactor when changing the reaction volume.

Reviewer’s Comment

  1. Can authors provide more details about the photoreactor design and construction, particularly in terms of preventing any potential interference from external factors such as ambient light?

Revisions and Comments

  • Thank you for the thoughtful question. To minimize the influence of ambient light, the space where the experiment was conducted was configured as a dark room; accordingly, the relevant information has been added to the text as follows.

“In addition, to minimize the influence of light other than that from the photoreactor and ultraviolet lamp used in the experiment, the space where the experiment was conducted was configured as a dark room.”

Reviewer’s Comment

  1. Were there any challenges or limitations in accurately determining the toluene concentration levels at different stages of the experiments, especially when dealing with trace amounts?

Revisions and Comments

  • Thank you for the insightful question. There were no special restrictions. However, because a ball flow meter was used rather than an MFC(Mass Flow Coltroller), a preliminary experiment was conducted to confirm initial concentration uniformity.

Reviewer’s Comment

  1. Considering that indoor environments can vary greatly in terms of temperature and humidity, how confident are authors that the observed results can be directly applied to different real-world scenarios?

Revisions and Comments

  • Thank you for the thoughtful question. To reflect the said content, information on the temperature and humidity values of apartment complexes in Korea, as confirmed in previous research, was added to the text as follows.

“In addition, the temperature and humidity values applied in the experiment were similar to those of apartment complexes in Korea, as confirmed through measurements by Hwang et al., assuming that actual indoor temperature and humidity values can be used.”

Related references have also been added.

Reviewer’s Comment

  1. Did authors observe any potential saturation effects in terms of the photocatalytic capacity of the TiO2 coatings, especially under high concentrations of toluene or extended exposure times?

Revisions and Comments

  • Thank you for the thoughtful question. Toluene, the pollutant used in this study, decomposes into CO2 and H2O through a reduction reaction caused by a photocatalyst. Saturation of substances such as existing NOx did not occur between experiments.

Reviewer’s Comment

  1. What strategies were employed to minimize the potential loss of TiO2 photocatalyst particles from the coating during the experimental setup and operation of the photoreactor?

Revisions and Comments

  • Thank you for the pertinent question. To minimize the potential loss of TiO2 photocatalyst particles, the first applied test specimen was stored in a vial to prevent scratching and peeling. The relevant information has been added to the main text as follows.

“To prevent damage to the TiO2 photocatalyst coating agent applied to the experimental specimen considering the repeated experiments, the experimental specimen was stored separately in a vial.”

Reviewer’s Comment

  1. Given the relationship between UV-A irradiance and reactor volume on the reduction efficiency, do authors envision any challenges in optimizing both parameters simultaneously for practical applications?

Revisions and Comments

  • Thank you for the insightful question. In real environments, there are limitations to ensuring smooth contact of contaminants with regards to the area where the photocatalyst is applied. Furthermore, in the case of indoors, there is a limit to the presence of UV-A, as mentioned in a previous comment. If these difficulties can be solved (for example, a closed schedule system and space utilization, etc.), an appropriate selection of the two parameters will seemingly be possible.

Reviewer’s Comment

  1. Could authors discuss the potential challenges of scaling up the application of TiO2 photocatalyst coatings to larger surfaces, such as entire rooms or buildings, and how the results might translate to such scenarios?

Revisions and Comments

  • Thank you for these thoughtful questions. Based on the results of this study, limitations regarding the application to actual buildings and corresponding avenues for future research were explored in the Discussion and Conclusion sections.

We look forward to hearing from you and would be happy to make further changes, if required.

Reviewer 2 Report

The manuscript, submitted to MDPI Materials by Y.W. Song et al. presents results on the gas-phase photocatalytic oxidation of toluene with commercial TiO2 photocatalyst, deposited on a glass substrate, which was characterised by SEM-EDX. The experiments were carried in a ISO22197-based flow reactor, and focussing on the effects of UV light intensity and the volume to the gas-flow cell.

Unfortunately, I cannot consider that the manuscript is suitable for publication in its current state. It lacks the structure of an academic text, provides great amount of superficial details about the equipment and apparatus used in the experiments, while on the other hand not enough information about crucial aspects of the sample preparation, and ultimately struggles to find a clear focus on why it is written, nor an useful discussion and interpretation of the results in contexts relatable to the topic of heterogeneous photocatalysis. The exposition is difficult to follow and the English is of distractingly, for a reader, poor quality, hence I think that it should be rejected, however, the authors should be encouraged to resubmit a re-written version. 

I want to underline, that this harsh criticism, should not be accepted by the authors as a discouragement, and it should be 100% clear that it is not an attack on their personal scientific merits. Au contraire, as of my experience in gas-phase photocatalysis, I think that there are some interesting points in the experiments, which can be underlined in a revised version and be of interest to the catalysis community. I also understand the effort required to construct the experimental setup, and even though the text is difficult to comprehend, and diluted by the extraneous information provided - I can even commend the authors for developing a logical experimental protocol, which suggest that their good intentions when the research was being planned.

So, I will take my time to provide some general suggestions and recommendations:

(1) The authors should consider to present their results in a context, which demonstrates novelty. Judging by the title, and the abstract, it seems as if the purpose of this work is to show that toluene can be removed by a TiO2 layer. 

The TiO2 material is commercial; the setup is according to the ISO standard, but the title of the standard is “ISO 22197-3:2019. Test method for air-purification performance of semiconducting photocatalytic materials — Part 3: Removal of toluene”. There are publications on toluene photooxidation since the 1986, hence, I am not certain that there is any need to further demonstrate this fact.

I can see that there are some sentences, like Lines 15-16 "the amount of ultraviolet-A (UV-A) light was varied depending on the lamp’s service life, a” and the paragraph in Lines 175-179 about the intensity reduction in UV lamps, suggesting that one of the ideas to study the UV intensity effect were, as a way to probably consider engineering aspects of photocatalytic air purifiers, as in how would their efficiency change with the degradation of the UV source, probably. However, this motivation is not clearly presented and difficult to find. 

Additionally, one of the aspects discussed in the paper are the effects of the volume of the reactor. Now, at a glance, in any real-world situation this is not a factor that can be varied, and a reader might be wondering why is it introduced. It is not made clear in the text, and the treatment of these results is quite superficial, and even in some case a bit erroneous. E.g., in lines 236-239 there is a claim that reducing the volume by 60% led to increased decomposition “owing to the reduction of amount of toluene that moved inside the reactor” - which is not true - for the same flow rate, the amount of toluene moving through the reactor will be the same - decreasing the volume will actually decrease the residence time of toluene in the residence time of toluene in the reaction volume, implying that the molecules passing over the TiO2 surface will be higher in number. Now, the fact that the authors measure higher conversion rate is interesting (and probably related to the smaller height of the laminar flow zone above the TiO2 surface, which would decrease the chances of mass-transfer effects // even though if I understand clearly there was a metal mesh in the flow, the reasoning of it is not clearly explained, but probably to induce turbulence).

I think that there are some good literature studies on both the effects of UV intensity, contaminant residence time in flow reactors and geometric aspects of the reactor, even providing mathematical models that the authors could use (would not list any here, to avoid endorsement of specific authors).

(2) The Introduction and Materials and methods sections need to be improved. The Introduction, probably, to a lesser extent, however, currently it is a list of findings by many studies and there is much attempt to tie them with the results discussed in the paper. E.g., there are some mentions about building materials with photocatalytic properties, some mentions about coatings of TiO2 to protect stainless steel from corrosion, etc. 

However, the Materials and methods section contains an overabundance of excessive information about the equipment used - especially Tables 2 & 4, information typically provided as a brief text outline, focusing on important details (e.g., I don’t think that a reader needs information on the electrical characteristics of the variable transformer used to control the UV tube). In contrast, very limited information is provided about the TiO2 coating - the paragraph in Lines 153-157 simply says that the TiO2 suspension was spray-coated and dried (To what temperature ? Were the 18 grams measured as the liquid dispersion, or was the final weight of the coating? What was the surface area dimensions on which they were coated ? Is there any possibilities for leftover binder to be present // judging by the EDS analysis probably not). 

(3) I also have some issues with the lack of discussion in the toluene mechanism of photo degradation and possible products. Maybe, given that the mechanism is well-elucidated in other publications, I am certain that at least a mention on the possible reaction products in this case is advisable. And while I don’t want to enforce requirements for any additional experiments to the authors - I believe that in future, they should attempt to provide at least some form of additional analysis of the resulting products - FTIR, GC/MS, even if these are unattainable - some form of CO2 measurement with NDIR-based sensor. The reason for this is - the Airwell plus gas analyser mentioned in Table 2 is probably non-selective (it even says on the screen TVOC) which puts under question the absolute treatment of the values provided, as ppm units.

While the English is legible and one can understand what the authors imply, I feel that improvements by the authors who are more fluent would be beneficial, given that the quality varies between the various sections of the paper. 

Author Response

Response to Reviewers

  1. Article Title: Indoor air pollutant (toluene) reduction according to ultraviolet-A irradiance and volume of reactor change in TiO2 photocatalyst reactor
  2. Submission Date: 21 August 2023
  3. Revisions and Comments: Reviewer 2

Thank you for taking the time to review our work and providing your valuable feedback. We appreciate your thoughtful consideration and kind response. We have carefully incorporated your suggestions into the revised manuscript, making sure to reflect your opinions as much as possible. To make it easier to identify the changes, we have marked them in red.

Reviewer’s Comment

  • The manuscript, submitted to MDPI Materials by Y.W. Song et al. presents results on the gas-phase photocatalytic oxidation of toluene with commercial TiO2 photocatalyst, deposited on a glass substrate, which was characterised by SEM-EDX. The experiments were carried in a ISO22197-based flow reactor, and focussing on the effects of UV light intensity and the volume to the gas-flow cell.

Unfortunately, I cannot consider that the manuscript is suitable for publication in its current state. It lacks the structure of an academic text, provides great amount of superficial details about the equipment and apparatus used in the experiments, while on the other hand not enough information about crucial aspects of the sample preparation, and ultimately struggles to find a clear focus on why it is written, nor an useful discussion and interpretation of the results in contexts relatable to the topic of heterogeneous photocatalysis. The exposition is difficult to follow and the English is of distractingly, for a reader, poor quality, hence I think that it should be rejected, however, the authors should be encouraged to resubmit a re-written version.

I want to underline, that this harsh criticism, should not be accepted by the authors as a discouragement, and it should be 100% clear that it is not an attack on their personal scientific merits. Au contraire, as of my experience in gas-phase photocatalysis, I think that there are some interesting points in the experiments, which can be underlined in a revised version and be of interest to the catalysis community. I also understand the effort required to construct the experimental setup, and even though the text is difficult to comprehend, and diluted by the extraneous information provided - I can even commend the authors for developing a logical experimental protocol, which suggest that their good intentions when the research was being planned.

So, I will take my time to provide some general suggestions and recommendations:

  • (1) The authors should consider to present their results in a context, which demonstrates novelty. Judging by the title, and the abstract, it seems as if the purpose of this work is to show that toluene can be removed by a TiO2

The TiO2 material is commercial; the setup is according to the ISO standard, but the title of the standard is “ISO 22197-3:2019. Test method for air-purification performance of semiconducting photocatalytic materials — Part 3: Removal of toluene”. There are publications on toluene photooxidation since the 1986, hence, I am not certain that there is any need to further demonstrate this fact.

I can see that there are some sentences, like Lines 15-16 "the amount of ultraviolet-A (UV-A) light was varied depending on the lamp’s service life, a” and the paragraph in Lines 175-179 about the intensity reduction in UV lamps, suggesting that one of the ideas to study the UV intensity effect were, as a way to probably consider engineering aspects of photocatalytic air purifiers, as in how would their efficiency change with the degradation of the UV source, probably. However, this motivation is not clearly presented and difficult to find.

Additionally, one of the aspects discussed in the paper are the effects of the volume of the reactor. Now, at a glance, in any real-world situation this is not a factor that can be varied, and a reader might be wondering why is it introduced. It is not made clear in the text, and the treatment of these results is quite superficial, and even in some case a bit erroneous. E.g., in lines 236-239 there is a claim that reducing the volume by 60% led to increased decomposition “owing to the reduction of amount of toluene that moved inside the reactor” - which is not true - for the same flow rate, the amount of toluene moving through the reactor will be the same - decreasing the volume will actually decrease the residence time of toluene in the residence time of toluene in the reaction volume, implying that the molecules passing over the TiO2 surface will be higher in number. Now, the fact that the authors measure higher conversion rate is interesting (and probably related to the smaller height of the laminar flow zone above the TiO2 surface, which would decrease the chances of mass-transfer effects // even though if I understand clearly there was a metal mesh in the flow, the reasoning of it is not clearly explained, but probably to induce turbulence).

I think that there are some good literature studies on both the effects of UV intensity, contaminant residence time in flow reactors and geometric aspects of the reactor, even providing mathematical models that the authors could use (would not list any here, to avoid endorsement of specific authors).

Revisions and Comments

  • Thank you for the extensive and thoughtful comments. The purpose and necessity of the research has been further elaborated in the Introduction section; moreover, related and relevant information has been added to Section 2.1 Overview. We hope that the additions correct the weaknesses that you have pointed out.

Reviewer’s Comment

(2) The Introduction and Materials and methods sections need to be improved. The Introduction, probably, to a lesser extent, however, currently it is a list of findings by many studies and there is much attempt to tie them with the results discussed in the paper. E.g., there are some mentions about building materials with photocatalytic properties, some mentions about coatings of TiO2 to protect stainless steel from corrosion, etc. However, the Materials and methods section contains an overabundance of excessive information about the equipment used - especially Tables 2 & 4, information typically provided as a brief text outline, focusing on important details (e.g., I don’t think that a reader needs information on the electrical characteristics of the variable transformer used to control the UV tube). In contrast, very limited information is provided about the TiO2 coating - the paragraph in Lines 153-157 simply says that the TiO2 suspension was spray-coated and dried (To what temperature ? Were the 18 grams measured as the liquid dispersion, or was the final weight of the coating? What was the surface area dimensions on which they were coated ? Is there any possibilities for leftover binder to be present // judging by the EDS analysis probably not).

Revisions and Comments

  • The Introduction and Materials and methods sections have been improved with regards to multiple aspects, as suggested by you and other reviewers. The purpose and necessity of the research has been further elaborated; moreover, some of the unnecessary equipment information has been excluded.

Reviewer’s Comment

(3) I also have some issues with the lack of discussion in the toluene mechanism of photo degradation and possible products. Maybe, given that the mechanism is well-elucidated in other publications, I am certain that at least a mention on the possible reaction products in this case is advisable. And while I don’t want to enforce requirements for any additional experiments to the authors - I believe that in future, they should attempt to provide at least some form of additional analysis of the resulting products - FTIR, GC/MS, even if these are unattainable - some form of CO2 measurement with NDIR-based sensor. The reason for this is - the Airwell plus gas analyser mentioned in Table 2 is probably non-selective (it even says on the screen TVOC) which puts under question the absolute treatment of the values provided, as ppm units.

Revisions and Comments

  • As suggested, we intend to perform additional experiments on reaction products as part of future research; this and other possible future avenues have been conveyed in the text as well. Finally, we have referenced the reaction products that you mentioned.

We look forward to hearing from you and would be happy to make further changes, if required.

Reviewer 3 Report

The article is interesting, but overall, the results and discussion could be more accurate. Please describe in more detail the obtained results. If we look at the amount of text in results and discussion - it seems small.

Additionally:

1) Abstract:

„. Toluene and TiO2 photocatalytic coating agents were used as representative indoor air pollutants.” – probably bad sentence construction – TiO2 was not used as an indoor pollutant.

2) 2. EXPERIMENTAL METHOD AND MATERIALS 104

2.1. Overview

“Some of the methods suggested in ISO 22197 were utilized for the experiment (International Organization for Standardization, 2011).- please write more about the mentioned methods, e.g.. list them in brackets.

3) In Figure 3. The author shows the spectral power distribution. Please provide the radiation intensity of the presented spectral range.

4) What does mean wheel sign in table 3, column 3: experimental application?

5) If possible please improve the quality of Figure 5. Please add in Figure 5 a description what is a and what is b.

Author Response

Response to Reviewers

  1. Article Title: Indoor air pollutant (toluene) reduction according to ultraviolet-A irradiance and volume of reactor change in TiO2 photocatalyst reactor
  2. Submission Date: 21 August 2023
  3. Revisions and Comments: Reviewer 3

Thank you for taking the time to review our work and providing your valuable feedback. We appreciate your thoughtful consideration and kind response. We have carefully incorporated your suggestions into the revised manuscript, making sure to reflect your opinions as much as possible. To make it easier to identify the changes, we have marked them in red.

Reviewer’s Comment

The article is interesting, but overall, the results and discussion could be more accurate. Please describe in more detail the obtained results. If we look at the amount of text in results and discussion - it seems small.

Revisions and Comments

As suggested by you and other reviewers, the Results and Discussion sections have been further strengthened with regards to multiple aspects.

Reviewer’s Comment

  • Abstract:

„. Toluene and TiO2 photocatalytic coating agents were used as representative indoor air pollutants.” – probably bad sentence construction – TiO2 was not used as an indoor pollutant.

Revisions and Comments

The Abstract has been revised as rightly pointed out by your comment.

“In the experiment, toluene, a representative indoor air pollutant, was removed using a coating agent containing TiO2 photocatalysts.”

Reviewer’s Comment

  • EXPERIMENTAL METHOD AND MATERIALS 104

2.1. Overview

“Some of the methods suggested in ISO 22197 were utilized for the experiment (International Organization for Standardization, 2011). - please write more about the mentioned methods, e.g.. list them in brackets.

Revisions and Comments

As suggested, more information has been added to the corresponding text.

“Some of the conditions suggested in ISO 22197 (UV-A irradiance of 10W/m2, pollutant flow rate of 3 lpm and concentration of 1.00 ppm, type of light-transmitting glass, etc.) …”

Reviewer’s Comment

  • In Figure 3. The author shows the spectral power distribution. Please provide the radiation intensity of the presented spectral range.

Revisions and Comments

As rightly suggested, more information has been added to the text.

“The wavelength range of the lamp can be determined through the power spectrum, and irradiance of approximately 20W/m2 is generated.”

Reviewer’s Comment

  • What does mean wheel sign in table 3, column 3: experimental application?

Revisions and Comments

There was an error in creating Table 3. Table 3 has been corrected and modified.

Reviewer’s Comment

  • If possible, please improve the quality of Figure 5. Please add in Figure 5 a description what is a and what is b.

Revisions and Comments

As suggested, Figure 5 has been improved in terms of quality. In addition, we have added information about (a) and (b) in Figure 5 to the main text.

“Figure 5 (a) illustrates the result of the FE-SEM and EDS mapping experiment before the TiO2 coating agent is applied, and (b) illustrates the measurement results of the experiment piece after the TiO2 coating agent is applied.”

We look forward to hearing from you and would be happy to make further changes, if required.

Round 2

Reviewer 1 Report

the revised manuscript can be accepted. 

Reviewer 2 Report

I believe that there is enough improvement for the manuscript to be considered for acceptance in its revised state. I would still keep on my recommendation to the authors, in future to consider more ways to plan research which expands, and contributes to the field of photocatalysis, and would nevertheless acknowledge their patience and effort in implementing the recommendations of the reviewers.

Need of some minor corrections is still required, but I am certain that this will be taken care of, through the feedback of the copyeditor. 

Reviewer 3 Report

In my opinion, the article can be published.